# Catalytical Performance of Heteroatom Doped and Undoped Carbon-Based Materials

Jahangir Alom [1,†], Md. Saif Hasan [1,†], Md. Asaduzaman [1,†], Mohammad Taufiq Alam [1], Dalel Belhaj [2], Raja Selvaraj [3], Md. Ashraf Hossain [4], Masoumeh Zargar [5,*] and Mohammad Boshir Ahmed [1,5,6,*]

1   Department of Applied Chemistry and Chemical Engineering, University of Rajshahi, Rajshahi 6205, Bangladesh
2   Department of Life Sciences, University of Sfax, FSS, Street of Soukra Km 3.5, BP 1171, Sfax CP 3000, Tunisia
3   Department of Chemical Engineering, Manipal Institute of Technology, Manipal Academy of Higher Education, Manipal 576104, Karnataka, India
4   Energy & Environmental Research Center, University of North Dakota, 15 North 23rd Street, Grand Forks, ND 58202, USA
5   School of Engineering, Edith Cowan University, Joondalup, WA 6027, Australia
6   School of Material Science and Engineering, Gwangju Institute of Science and Technology, Gwangju 61005, Republic of Korea
*   Correspondence: m.zargar@ecu.edu.au (M.Z.); mohammad.ahmed@gist.ac.kr (M.B.A.)
†   These authors contributed equally to this work.

**Abstract:** Developing cost-effective, eco-friendly, efficient, stable, and unique catalytic systems remains a crucial issue in catalysis. Due to their superior physicochemical and electrochemical properties, exceptional structural characteristics, environmental friendliness, economic productivity, minimal energy demand, and abundant supply, a significant amount of research has been devoted to the development of various doped carbon materials as efficient catalysts. In addition, carbon-based materials (CBMs) with specified doping have lately become significant members of the carbon group, showing promise for a broad range of uses (e.g., catalysis, environmental remediation, critical chemical production, and energy conversion and storage). This study will, therefore, pay attention to the function of heteroatom-based doped and undoped CBMs for catalytical applications and discuss the underlying chemistries of catalysis. According to the findings, doping CBMs may greatly improve their catalytic activity, and heteroatom-doped CBMs may be a promising option for further metal doping to attach them to an appropriate place. This paper also covers the potential applications of both doped and undoped CBMs in the future.

**Keywords:** doping carbon materials catalyst; OER; HER; methanol oxidation; photocatalysis

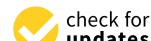



## 1. Introduction

The incessant growth of energy demand and environmental pollution are the most concerning current issues worldwide. The inadequate supply of fossil fuels for the rising global population and the harmful pollution from fuel-burn discharges have compelled researchers to find a new, sustainable way to produce energy by utilizing renewable energy sources [1,2]. In recent years, some environmentally friendly technologies such as water splitting, fuel cells, and metal-air batteries have come to the limelight by showing their promising energy conversion and storage efficiency as well as their sustainable performances [3]. The major roadblock to the sustainable grid-scale development of these renewable energy technologies is the requirement of noble-metal catalysts for these reactions (e.g., Pt, Ru, Pd, Ir, etc.), which are highly expensive. Due to the scarcity and high cost of these noble metal catalysts, recent research has focused on the reduction of their gross amount in electrochemical cells. As a part of these alternative approaches, different advanced carbon-based materials (CBMs), such as graphene, graphene oxides (GO), graphitic

carbon nitrides (GCN), carbon nanotubes (CNT), etc., have been suggested as promising supporting materials for the conventional metal (oxide) catalysts. These carbon-based supports have gained considerable attention from researchers during the past few years due to their several advantages, such as large surface area, good electrochemical stability, and low cost of synthesis [4]. However, one major drawback of these supports is their surface inertness compared to metal catalysts. It is difficult to deposit the catalytic nanoparticles on the inert surface of these supporting materials. Thus, the poor catalyst-support interaction restricts the development of a uniformly dispersed, efficient, and durable ideal catalyst.

Doping CBMs with various heteroatoms (e.g., N, S, P, B, etc.) is a promising solution to this problem [5]. These heteroatoms are composed of a different number of electrons and electronic configurations than that of *C*-atoms. Therefore, carbon materials must be chemically modified when they are doped with these atoms. The tuned chemical and electronic properties of the doped CBMs help the uniform dispersion of the catalysts on the support and increase catalyst stability [6]. This also helps to enhance the catalytic activity of the metal single-atom catalysts (SACs) [7]. Furthermore, heteroatom-doped CBMs are reported as promising metal-free catalysts for water-splitting and oxygen reduction reactions (ORR) in fuel cells [8]. Moreover, some of these metal-free catalysts increased the power conversion efficiency of the fuel cells at a relatively lower cost compared to the noble-metal catalyst-based fuel cells. Therefore, the heteroatom doping of CBMs is a topic of great importance for sustainable energy production and storage technologies.

Some advanced CBMs, such as GCN, graphene, and GOs, are also used as metal-free green catalysts for the photochemical degradation of organic pollutants from water (Figure 1). These materials have a narrow or zero band gap between their valence band and conduction band, and thus, are capable of producing pairs of photogenerated electrons and holes by using renewable energies from sunlight [9]. However, the slower rate of charge transfer and the charge-carrier recombination in pure CBMs compared to conventional inorganic catalysts (such as $TiO_2$, ZnO, and $SnO_2$ nanoparticles) limits their photocatalytic performances. To overcome these limitations, various heteroatoms are doped into these photo-active CBMs. It has been found in many studies that the heteroatom doping of these materials helps to increase their light absorption capacity [10], promotes charge separation [11], differs the electron/hole recombination rate [12], and increases the rate of charge transfer. Therefore, the heteroatom-doped CBMs are promising photocatalysts for the decomposition of organic pollutants.

Although some existing reviews are available on the photocatalytic and electrocatalytic application of various doped CBMs, most of them are based on some specific CBM (e.g., carbon dots, graphene), specific element/s doping (e.g., doping with nitrogen), or some specific reactions (e.g., HER, ORR). For instance, recently, Salinas-Torres et al. [13] reviewed the application of nitrogen-doped CBMs for hydrogen-generation reactions. However, the effect of doping with other heteroatoms was not discussed in the article. Zhang et al. [14] studied the recent advances in the application of doped carbon dots for electrocatalytic reactions. However, the study was confined to carbon dots only. Other heteroatom-doped CBMs, such as graphene, GCN, and CNTs, were not reviewed. Hu et al. [15] reviewed the doped CBMs for various electrocatalysis reactions. The photocatalytic application of such materials was not discussed in the study. Other review articles based on the N-doped CBMs for ORR [16], transition metals doped CBMs for HER [17], catalytic applications of metal-N-doped CBMs [18], and electrocatalytic applications of non-N-doped CBMs [19] are also available. However, to the best of the authors' knowledge, there is no review in the literature where the role and underlying chemistry of heteroatom doping on the electrocatalytic and photocatalytic behavior of different CBMs (e.g., graphene, carbon dots, GCN, CNTs, etc.) have been discussed together. In addition, the comparisons between the catalytic performances of the doped CBMs and the origin of their performance deviations have not yet been reviewed collectively in any study (Figure 1).

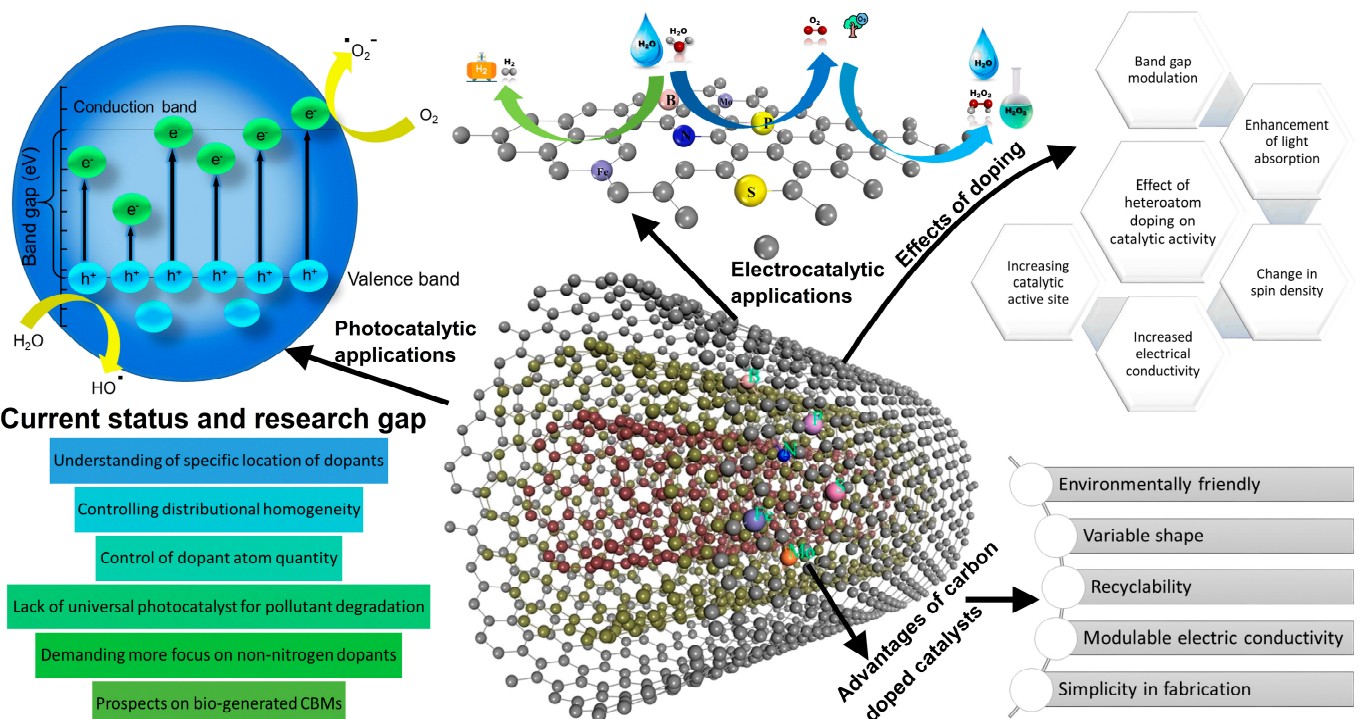

**Figure 1.** Current status and advances of doped carbon materials for various applications.

In our study, we have reviewed the photocatalytic and electrocatalytic application of various heteroatom-doped CBMs in the reactions, such as hydrogen evolution reaction (HER), oxygen evolution reaction (OER), ORR, methanol oxidation, and pollutants catalysis. In addition, the effects of various dopant elements on the performance of these CBMs and the underlying chemistries of these effects have also been discussed in detail.

## 2. CBMs as Catalysts in Energy Conversion and Storage

### 2.1. Oxygen Evolution Reaction

2.1.1. Role of CBMs in the OER

OER is a limiting process in the production of molecular oxygen by chemical reactions, such as water electrolysis into oxygen and hydrogen, water oxidation in oxygenic photo-synthesis, and electrocatalytic oxygen evolution from oxides and oxoacids. It can occur either in an alkaline, acidic or neutral environment, as illustrated in Figure 2 [20,21]. The suggested OER mechanisms use the same intermediates (i.e., OH*, OOH*, and O*) as the ORR for both environments. However, the first step for OER is the formation of absorbed OH* on the catalysts with the first electron transfer (Equations (1) and (5)). The transition of OH* to O* is the next step (Equations (2) and (6)). The third step is to convert O* to OOH* by combining it with another $H_2O$ molecule or $OH^-$ (Equations (3) and (7)). The final stage is to release $O_2$ (Equations (4) and (8)).

In the acidic medium, the OER reaction mechanism can be written as [22],

$$H_2O\ (l) + * \ \rightarrow OH^* + H^+ + e^- \tag{1}$$

$$OH^* \ \rightarrow O^* + H^+ + e^- \tag{2}$$

$$O^* + H_2O\ (l) \ \rightarrow OOH^* + H^+ + e^- \tag{3}$$

$$OOH^* \ \rightarrow * + O_2(g) + H^+ + e^- \tag{4}$$

In an alkaline medium, the OER reaction mechanism can be written as [23],

$$OH^- + * \ \rightarrow OH^* + e^- \tag{5}$$

$$OH^* + OH^- \rightarrow O^* + H_2O(l) + e^- \tag{6}$$

$$O^* + OH^- \rightarrow OOH^* + e^- \tag{7}$$

$$OOH^* + OH^- \rightarrow * + O_2(g) + H_2O(l) + e^- \tag{8}$$

where (*l*) stands for the liquid phase; (*g*) for the gas phase; * for the active site on the catalyst; and O*, OH*, and OOH* as adsorbate intermediates.

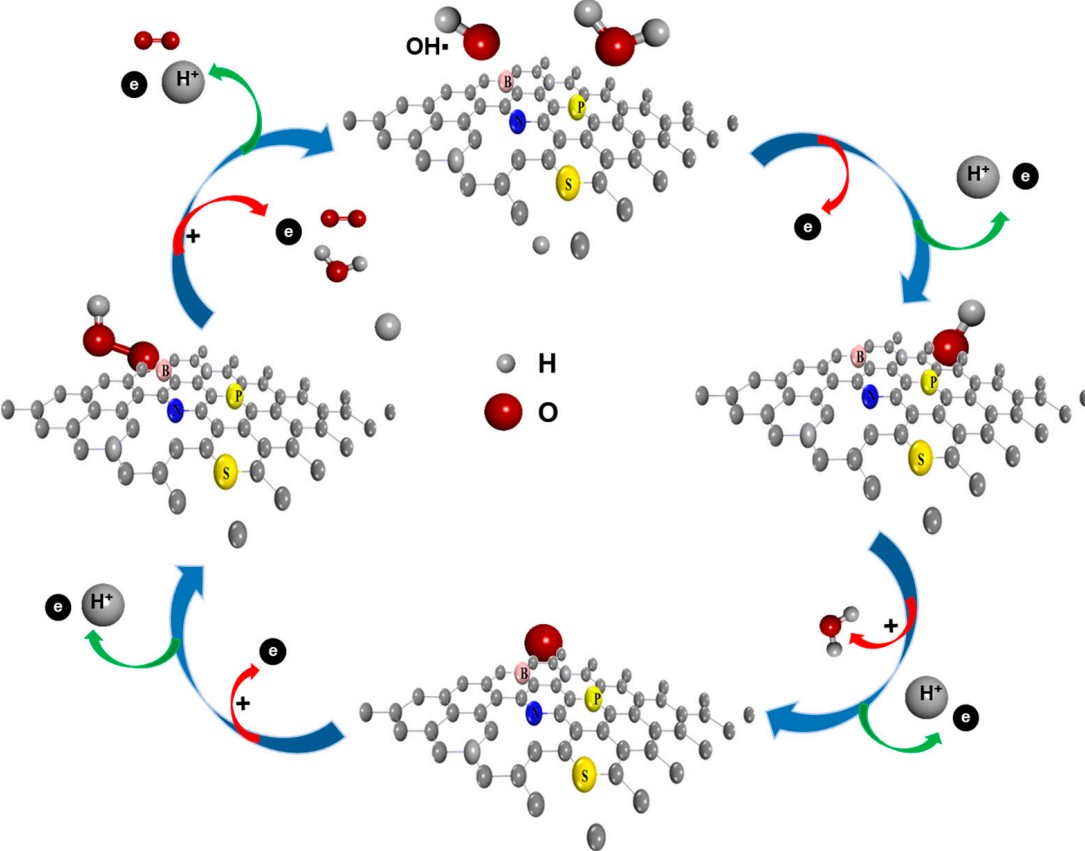

**Figure 2.** Mechanisms of OER in both basic (through the red route) and acidic (via the green route) environments.

Each stage involves the release of a single electron. Therefore, OER involves a four-electron transfer pathway and is a much more complicated and energy-consuming process [23–27]. The primary challenges of these multi-electron transfer systems are the high overpotential and the energy barrier of the rate-determining phase (Figure 3a). The essential overpotential of the reaction might be avoided by stabilizing the HOO* concerning HO* intermediate on the catalyst surface. Catalytically, precious metal complexes have a good size-to-performance ratio, but they are also volatile organic compounds, costly, and potentially toxic. Graphene, a 2D material derived from carbon, was introduced in the latter part of the twenty-first century to alter, help, or replace these metal-based electrocatalysts [25]. Due to their adjustable molecular structures, abundance, and excellent tolerance to acidic and alkaline environments, many carbon-based materials provide distinct advantages for targeted catalysis. Low-dimensional carbon materials have recently demonstrated their potential as metal-free catalysts in energy-related electrocatalytic OERs [26]. For instance, graphene is the most effective alternative catalyst due to its sp$^2$ carbon atoms being organized in a honeycomb shape with a π-π conjugation as well as its great mechanical strength and electrical conductivity [27].

### 2.1.2. Effect of Heteroatom Doping on OER

The OER process can be efficiently catalyzed by using CBMS catalysts. Despite their advantageous catalytic size performance ratio, precious metal complexes are unstable, costly, and toxic. However, pure carbon materials are often inactive for OER because of the absence of active sites for the adsorption and stimulation of $O_2$ and/or intermediates [28–32]. Therefore, doping heteroatoms (e.g., N, S, P, B, and transition metals) on the surface of carbon-based materials causes an electron and spin density distortion, which results in increased catalytic activity towards OER [24,29,30]. For instance, single elements, particularly N-doped carbon material, have been extensively investigated as bifunctional electrocatalysts for OER [24,31]. The N-atom doped CBMs show an OER over the potential of 380 mV (current density of 10 mAcm$^{-2}$ at p$^H$ 13), which are values equivalent to those of iridium oxide and cobalt oxide as well as the platinum catalyst (Figure 3b) [24]. In addition, as compared to the noble-metal catalyst $IrO_2$ (η = 350 mV and Tafel slope ~89 mVdec$^{-1}$), the N-doped mesoporous carbon nanotube/carbon nanosheet hybrid had exceptional overpotential (i.e., η = 320 mV at 10 mAcm$^{-2}$ in an alkaline medium, e.g., 0.1 M KOH) and Tafel slope of ~55 mVdec$^{-1}$ with a minimal onset potential of 1.50 V, vs. RHE for OER (Figure 3c,d) [32]. Moreover, the N-atom doped mesoporous graphene catalyst has a lower OER overpotential (324 mV at 10 mA cm$^{-2}$) and a small Tafel slope of 67 mV dec$^{-1}$, which is lower than most of the previously disclosed non-noble metal oxides and their hybrids and also comparable to the OER standard of a noble metal or metal oxide catalysts (Ru, $RuO_2$, Ir, and $IrO_2$) [29]. In addition, doping multiple heteroatoms is more likely to provide more active sites than doping single heteroatoms, resulting in higher catalytic activity for OER. To explain, an N/P co-doped graphene/carbon nanosheet catalyst presented better electrocatalytic performance with an onset potential of 1.57 V vs. RHE and a low overpotential of 319 mV at 10 mAcm$^{-2}$ (Tafel slope of ~70 mVdec$^{-1}$), which is lower than a mono heteroatom, e.g., N/P doped graphene, as well as a noble metal catalyst, e.g., $RuO_2$, Pt/C [33] (Figure 3e). Another research group observed that an N-P co-doped graphene catalyst had a minimum OER overpotential of 390 mV, which is lower than the best catalyst identified theoretically (420 mV for OER in $RuO_2$) [23]. This result is represented in Figure 3f, which is called the volcano plot. Moreover, incorporating B into an N-doped carbon electrocatalyst for boosting OER performance presented a superior onset potential of 1.46 V vs. RHE and over the potential of 270 mV at 10 mAcm$^{-2}$, which was lower than commercial 5 wt.% Ru/C (onset potential of 1.48 V vs. RHE and η = 275 mV) [34]. In addition, Zhao et al. [35] revealed that S and N co-doped graphene/CNTs exhibit a higher negative onset potential of 0.56 V in 0.1 M KOH (η = 510 mV) and a lower Tafel slope of 103 mV dec$^{-1}$, indicating faster OER kinetics compared to single N-doped graphene/CNTs (0.65 V, 285 mV dec$^{-1}$) and commercial Pt/C catalyst (η = 760 mV).

Nevertheless, a combination of non-metal and transition metal doping in OER increases the catalytic activity of carbon-based materials [36]. Examples include metal and N co-doped CBMs such as Fe-N co-doped CBMs and Co-N co-doped CBMs, which exhibited low overpotentials of 360 mV and 380 mV at 10 mAcm$^{-2}$, respectively, which are comparable with that of $IrO_2$/C (η = 370 mV at 10 mAcm$^{-2}$) [23]. Moreover, Fe on an N/S co-doped C catalyst generated reasonably good activity for OER with an overpotential of 370 mV at 10 mAcm$^{-2}$ in an alkaline medium and a small Tafel slope of ~82 mVdec$^{-1}$, which is comparable with a Pt/C catalyst [37]. The ranges of overpotentials at 10 mAcm$^{-2}$ acquired from the literature on different techniques for OER are summarized in Table 1. It can be concluded that heteroatom-doped carbon catalysts have better OER activity than metal-based catalysts. Multiple heteroatoms doped in CBMs, on the other hand, have demonstrated high OER catalytic activity. Co-doping with transition metals in carbon frameworks does not appear to work on OER, as evidenced by the relatively high overpotentials.

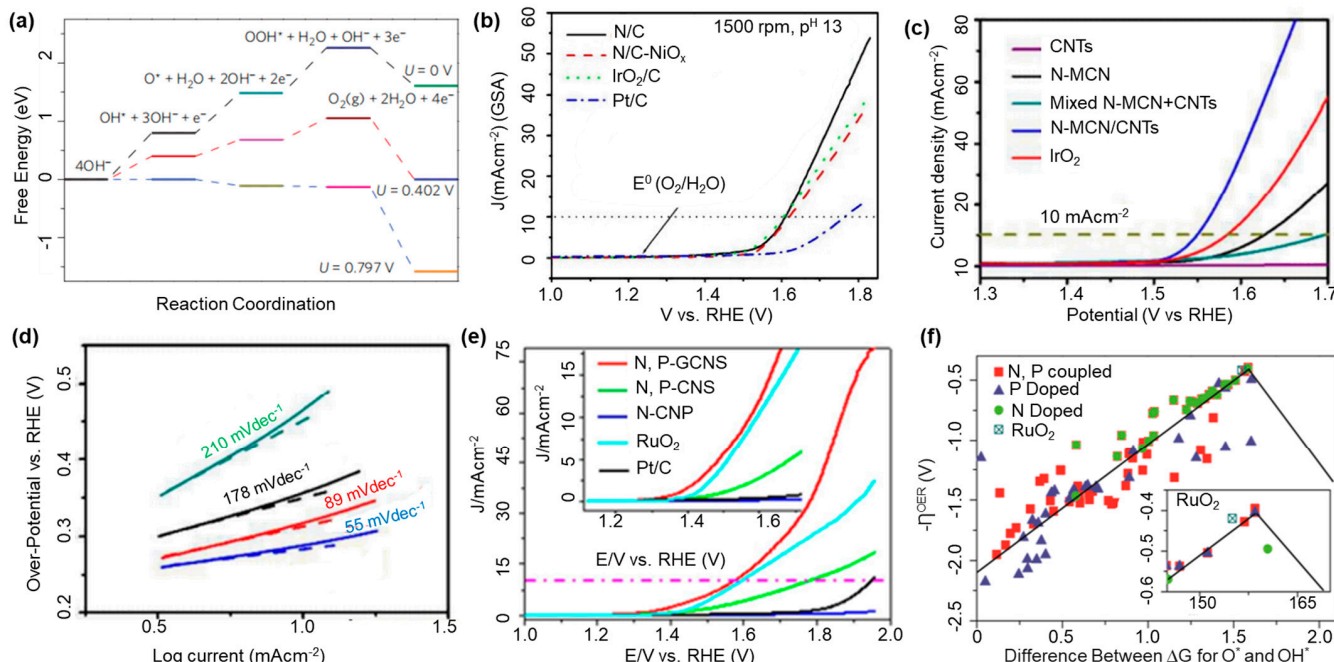

**Figure 3.** (**a**) Schematic energy profiles for the OER pathway; modified from [23]. (**b**) Compares the OER activities of N-doped carbon material with other different catalysts; reproduced with permission from [24], Copyright@2004, American Chemical Society. (**c**) LSVs and (**d**) Tafel plots of different catalysts; reproduced with permission from [32], Copyright@2015. (**e**) Comparison of the OER activity of various catalysts; reproduced with permission from [33], Copyright@2015, American Chemical Society. (**f**) Overpotential η vs. O* adsorption energy (volcano plot) and the difference between the adsorption energy of O* and OH* for N-doped, P-doped, and N, P-doped graphene OERs, modified from [23].

**Table 1.** Summary of overpotentials and Tafel slope toward OER for non-metal CBMs.

| CBMs | | Overpotential at 10 mAcm$^{-2}$ in 0.1 M KOH (mV) | Tafel Slope mVdec$^{-1}$ | Ref. |
|---|---|---|---|---|
| N doped CBMs | N-doped mesoporous carbon nanosheet/carbon nanotube hybrid | 320 | 55 | [22–24,29,38–40] |
| | N-doped mesoporous carbon nanosheet | 400 | 178 | |
| | N-doped Carbon Nanomaterials | 380 | - | |
| | Mixed N-doped mesoporous carbon nanosheet + carbon nanotube | 470 | 210 | |
| | N-doped mesoporous graphene | 324 | 67 | |
| | N-MWCNTs | 320 (1 M NaOH) | 68 | |
| | N-Graphene | 410 | - | |
| | N-CNTs | 390 | - | |
| | N-Graphene Nanoribbon | 405 | - | |
| | N-graphene/CNT hybrid | <110 | 83 | |
| | N-Graphene Nanoribbon | 360 (1 M KOH) | 47 | [38] |
| | N-Carbon nanosheet | 410 | 142 | [41] |
| | N- Graphitic mesoporous C$_3$N$_4$ | 376 | 52.4 | [42] |
| P-doped CBMs | P-doped graphitic carbon nitride grown on carbon-fiber paper | 391 | 61.6 | [23,43,44] |
| | P-graphene | 490 | - | |
| | P-Graphene | 330 (1 M KOH) | 62 | |
| S-doped CBMs | S-CNTs | 350 (1 M KOH) | 95 | [45] |
| P-S co-doped CBMs | P, S-doped carbon nitride sponge | 330 | 64 | [46] |
| S-N co-doped CBMs | N/S-CNTs | 351(1 M KOH) | 56 | [35,47] |
| | NS-graphene/CNT | 510 | 103 | |
| | N-S doped graphitic sheet | 230 | 71 | [48] |
| -F co-doped CBMs | N-F co-doped carbon black | - | 69 | [49] |
| B-N co-doped CBMs | B/N-C | 270 (1 M KOH) | 100 | [34] |

## 2.2. Oxygen Reduction Reaction (ORR)

### 2.2.1. Role of CBMs in an ORR

Highly efficient and stable electrocatalysts are required to speed up the ORR at the cathode in proton exchange membrane fuel cells (PEMFC) or metal-air batteries (MABs). When electrons reduce $O_2$ molecules at the cathode of PEMFCs or MABs during the discharging process, an ORR takes place (Figure 4). However, electrochemically breaking the O=O bond is challenging due to its high bond energy of 498 kJmol$^{-1}$ [50]. So, electrocatalysts are required to reduce the energy barrier during bond activation and breaking (Figure 5a). Due to the many adsorption/desorption and reaction mechanisms involving separate O-containing intermediates (e.g., OOH*, OH*, and O*), the ORR at the cathode in PEMFCs is about six orders of magnitude slower compared to hydrogen oxidation at the anode in an aqueous medium [51]. Hence, the cathode's catalyst consumption is often 10 times that of the anode [52]. The scarcity and expensive cost of Pt-based electrocatalysts limit the widespread use of MABs or PEMFCs. Due to their large surface region, high conductivity, variable shape, simplicity of fabrication, and economic feasibility, metal-free CBMs are a unique sort of catalyst that has the potential to replace Pt in effectively catalyzing the ORR in fuel cells [53]. While significant progress has been made in creating improved carbon compounds as very stable and durable catalysts, the catalytic processes of CBMs remain unknown.

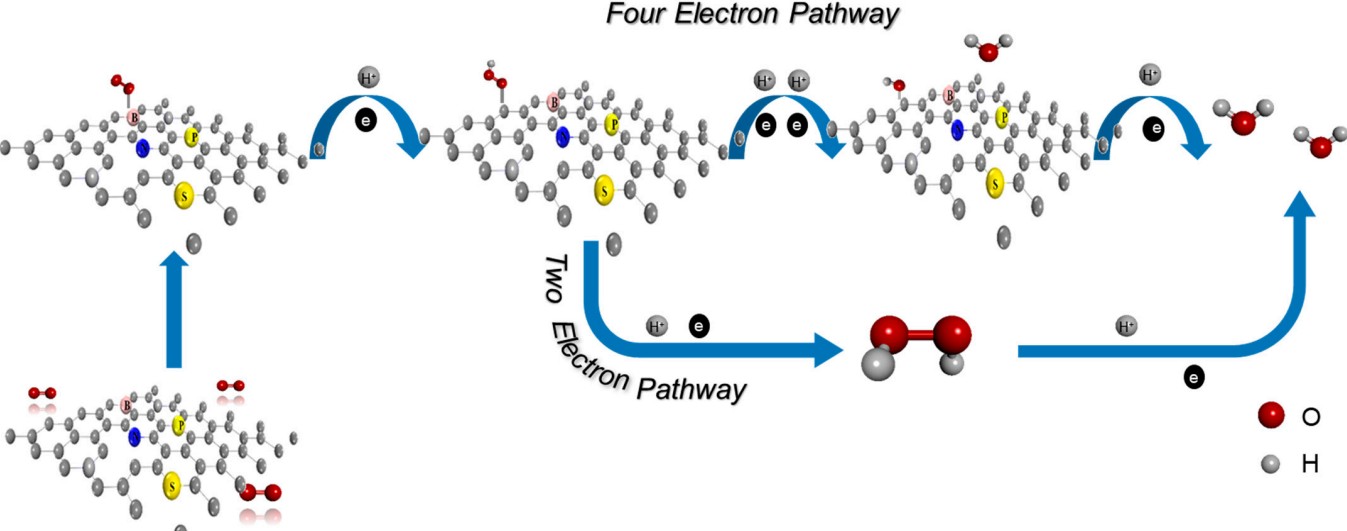

**Figure 4.** Mechanism of ORR in CBMs catalyst.

In general, ORR is a multistep electrochemical process (2e pathway) that can involve the formation of $H_2O_2$ (in acidic media) or $HO_2^-$ (in alkaline media) as the intermediate specie, or it can proceed more efficiently via a four-electron (4e) process involving the direct reduction of $O_2$ into $H_2O$ (in acidic media) or $OH^-$ (in alkaline media) to combine with a proton into water.

The reactions in an acidic (Equation (9)) or an alkaline (Equation (10)) electrolyte for a straight four-electron route are as follows:

$$O_2 + 4H^+ + 4e^- \rightarrow 2H_2O \tag{9}$$

$$O_2 + 2H_2O + 4e^- \rightarrow 4OH^- \tag{10}$$

For an indirect two-electron pathway, the reactions in alkaline (Equations (11) and (12)) or acidic (Equations (13) and (14)) electrolytes are as follows:

$$O_2 + H_2O + 2e^- \rightarrow HO_2^- + OH^- \tag{11}$$

$$HO_2^- + H_2O + 2e^- \rightarrow 3OH^- \tag{12}$$

$$O_2 + 2H^+ + 2e^- \rightarrow H_2O_2 \tag{13}$$

$$H_2O_2 + 2H^+ + 2e^- \rightarrow 2H_2O \tag{14}$$

Both methods depend crucially on the oxygen ($O_2$) adsorption mode and the dissociation restriction of the O-O bond on the catalyst's contact. In contrast, reaction-free energy was commonly used as a descriptor to evaluate the catalytic activity or specificity of novel electrocatalysts due to their rapid development. Two putative ORR processes are presented in the electrochemical system created by Nrskov et al. [54] an additive mechanism that involves a HOO* species and a direct $O_2$ dissociation mechanism in an acid or alkaline electrolyte, which are represented in Figure 4.

### 2.2.2. Effect of Heteroatom Doping on ORR

Pristine CMBs have lower performance compared to doped carbon materials. Metal-free carbon materials, particularly after doping heteroatoms or generating defects, displayed exceptional ORR activity and were sometimes comparable with commercial platinum-based catalysts (Table 2). For example, nano-forests of N-doped carbon nanotubes outperformed commercially available Pt/C electrodes in ORR [8]. Similarly, improved catalytic performance for N/B-doped and S-doped was observed for ORR due to the redistribution of the charge density and the spin density of adjacent C atoms [5,55]. Li et al. [32] evaluated CNT, nanoporous carbon-nanofiber film (N-CNF), N-doped mesoporous carbon nanosheet, and $IrO_2$ for ORR reactions. They found an excellent ORR activity exhibition by CBMs compared to $IrO_2$. Similarly, Jiao et al. [56] studied the activity of ORR on graphene with various dopants such as N, B, S, and P. Among them, N and B showed the best ORR performance with the lowest free energy change at the equilibrium potential U0 = 0.455 V. For both dopants, adsorption of $O_2$ on graphene is an endothermic reaction with positive free energy ($\Delta G = 0.70$ eV). The formation of *O and *OH is exothermic ($\Delta G = 0.54$ eV, $\Delta G = 0.25$ eV and $\Delta G = 0.58$ eV, $\Delta G = 0.30$ eV, and $\Delta G = 0.58$ eV respectively) while the formation of OH$^-$ has a slightly positive energy which is easily surmountable [56].

Moreover, based on diffusion limiting current density ($J_L$) and onset potential ($E_{onset}$), N, S, B, and P-doped CBMs also improve ORR activity compared to the undoped counterparts. N-doped CNTs, for example, showed a higher limiting current density of three times compared to undoped CNTs, which indicates a significant increase in catalytic activity (Figure 5b) [57]. Furthermore, compared to the non-doped graphene, the reduction potential of S-doped graphene shows a positive shift of 40 and 60 mV and higher current density, which indicates an electrocatalytic site towards ORR [58]. Similarly, compared to the pristine CBMs, B-doped CNTs or graphene and P-doped graphene also demonstrated superior ORR activity, but it was inferior to that of the commercial Pt/C, unlike N-doped CBMs [55,59]. In contrast, F-doped Carbon Blacks with higher electronegativity ($X_F = 3.98$) demonstrated unique catalytic activity, with higher positive $E_{onset}$ and half-wave potential ($E_{1/2}$) values than Pt/C [60].

On the other hand, co-doped CBMs such as N with another heteroatom exhibit better electrocatalytic activity toward ORR than the single heteroatom-doped CBMS. N, B co-doped graphene, for example, demonstrated better electrocatalytic ORR performance than the commercial Pt/C (Figure 5c) [61]. A group of researchers determined the best ORR catalyst by calculating the overpotential, η, a key measure of catalytic activity, for each active site on N and P co-doped graphene structures in alkaline environments [23]. The result, depicted in a "volcano plot" showed that the N and P co-doping generated synergetic effects, leading to a minimum ORR overpotential of 0.44 V, outperforming the best catalyst (0.45 V for ORR on Pt) (Figure 5d). The N and P co-doping proved to be a catalyst with superior electrocatalytic activity for ORR. In addition, N, S co-doped graphene demonstrated more positive $E_{onset}$ (−0.06 V vs. Ag/AgCl) than that of N-graphene, S-graphene, or pristine graphene, which was comparable to commercial Pt/C (−0.03 V vs. Ag/AgCl) [62]. However, N, S, and P co-doped CBMs with porous structures carbonized

from metal-organic frameworks also exhibited similar performances to that of commercial Pt/C e [63]. At low overpotential, N, S, and P co-doped graphene showed a Tafel slope of 72 mVdec$^{-1}$, which was closer to that of commercial Pt/C (69 mVdec$^{-1}$) [64]. The low Tafel slope indicates a faster catalytic activity toward ORR in the N, S, and P co-doped graphene catalyst surfaces.

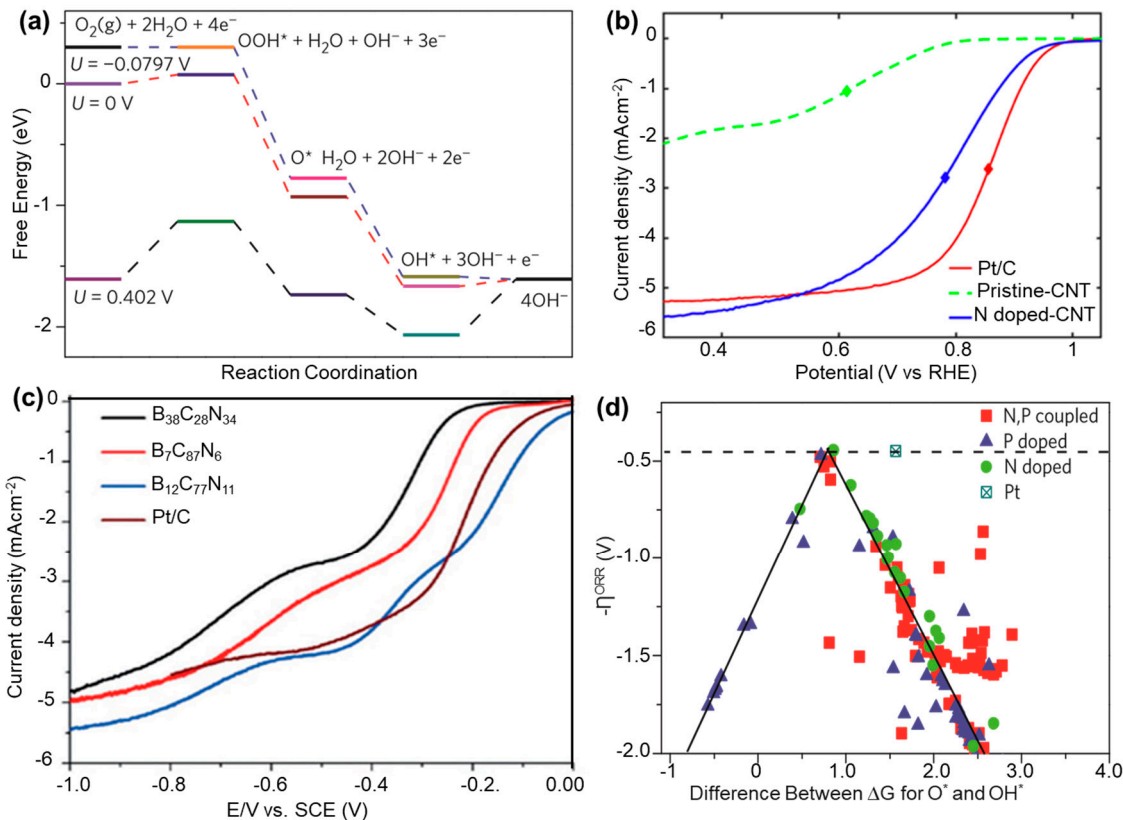

**Figure 5.** (**a**) Schematic energy profiles for the ORR pathway; modified from [23]. (**b**) LSV curves of Pt/C, pristine-CNT, and N-doped CNT catalyst samples at a rotation speed of 1600 rpm; reproduced with permission from [57], Copyright@2020, American Chemical Society. (**c**) LSV curves of ORR on BCN graphene with different compositions in oxygen-saturated 0.1 m KOH solution at 10 mVs$^{-1}$ and compared with the commercial Pt/C electrocatalyst; reproduced with permission from [61], Copyright@2012, John Wiley and Sons License. (**d**) ORR volcano plots of overpotential η versus adsorption energy of O* and the difference between the adsorption energy of O* and OH*, respectively, for N-doped, P-doped, and N, P-doped graphene; modified from [23].

**Table 2.** Comparison of specific performance of the heteroatom-doped CBMs ORR catalysts.

| Electrocatalyst for ORR | | No. of Electrons Transferred | On-Set Potential ($E_{onset}$) (V) | Half-Wave Potential($E_{1/2}$) (V) | Limiting Current Density (mA/cm²) | Ref. |
|---|---|---|---|---|---|---|
| | Pt/C | 3.99 | 1.00 | 0.86 | −5.24 | |
| | pristine CNT | - | 0.80 | 0.61 | −2.62 | [57] |
| N-doped CBMs | N-CNT | 3.92 | 0.96 | 0.78 | −5.82 | |
| | N-mesoporous carbon | 4.02 | 0.949 | lesser as compared to Pt/C | ~−4.9 | [65] |
| | Pt/C | - | 0.95 | - | - | |
| | Pristine Graphene | - | 0.82 | - | - | [59] |
| P-doped CBMs | P-graphene | 3.8 | 0.92 | - | −3.63 | |
| | P-graphene/carbon black | 3.8 | 0.92 | - | −4.18 | |
| N, P-co-doped CBMs | N, P doped graphene | | 0.88 | 0.774 | - | [64] |
| | N, P-doped mesoporous carbon | ~4.0 | 0.94 | 0.85 | comparable to that of Pt/C | [23] |
| N, O-doped mesoporous carbon | | 3.78–4 | 0.5–0.7 | - | ~−5.3 | [66] |
| N, S, P co-doped graphene | | >3.8 | 0.96 | 0.857 | - | [64] |

All these electrocatalysts were tested in 0.1 M KOH electrolyte solution at 1600 rpm.

### 2.3. Hydrogen Evolution Reaction (HER)

2.3.1. Role of CBMs in a HER

Hydrogen, as a clean source of renewable energy, may be the key to addressing challenges of energy sustainability as well as the environment. It is possible to produce hydrogen at room temperature by a process called water electrolysis, which is one of the most efficient techniques. As compared to other catalysts, Pt and its alloys have the greatest electrocatalytic activity and a very high exchange current density [67]. However, its high cost and lack of available sources provide significant barriers to large-scale hydrogen generation [68]. As a result, significant efforts have been devoted to producing non-Pt electrocatalysts, such as metal alloys [69], carbides [68], phosphides [70], borides [71], sulfides, and nitrides [72]. Along with significant research efforts in generating non-Pt electrochemical catalysts, a novel class of catalysts based on carbon materials has been discovered, which might drastically cut costs while providing excellent efficiency and stability. When compared to metal-based catalysts, carbon-based, metal-free alternatives have been shown to offer several benefits, including high electrical conductivities, flexible molecular architectures, abundance, and good tolerance to acidic/alkaline conditions. The current availability of carbon materials opens new avenues for creating sophisticated metal-free catalysts with outstanding catalytic activity.

Knowledge of the HER mechanism is vital to building the next generation of HER catalysts. HER occurs through the reduction of protons ($H^+$) or water ($H_2O$) molecules, followed by the generation of gaseous hydrogen depending on the pH of the electrolyte solution (Figure 6).

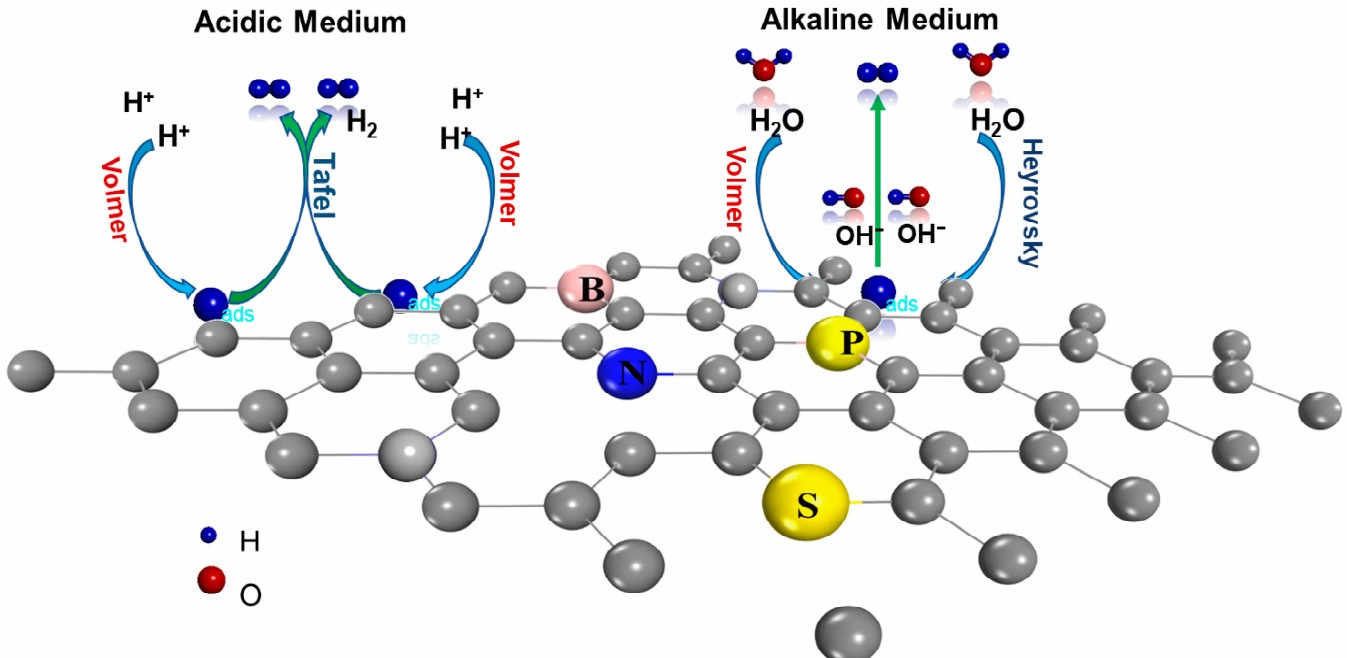

**Figure 6.** HER mechanism in CBMs catalyst.

In acidic media, the overall HER follows the general Equation (15).

$$2H^+ \text{ (aq)} + 2e^- \rightarrow H_2 \text{ (g)} \tag{15}$$

In alkaline media, the overall HER reaction is described in Equation (16).

$$2H_2O \text{ (l)} + 2e^- \rightarrow 2OH^- \text{ (aq)} + H_2 \text{ (g)} \tag{16}$$

The following sections describe the progress made in the development of metal-free heteroatom-doped carbon-based catalysts for the HER.

2.3.2. Effect of Heteroatom Doping on HER

Though pristine carbon materials show very poor catalytic activity, a slight chemical modification of the material via doping of some heteroatoms can significantly increase the electrocatalytic nature of CBMs in the case of HER [73]. Because heteroatoms differ in size and electron negativity from carbon atoms, the introduction of heteroatoms may cause electron modulation to change the charge distribution and electronic properties of carbon skeletons, affecting their interaction with hydrogen intermediates and, ultimately, their electrocatalytic activities for the HER [74]. This, together with the doping-induced flaws, has the potential to modify the chemical activity of carbon materials resulting in increased electrocatalytic activity, notably in water splitting.

For example, O-doped CNTs showed an onset over the potential of 100 mV, a Tafel slope of 82 mV dec$^{-1}$, and required an overpotential of 220 mV to reach 10 mAcm$^{-2}$ [75]. A similar result is also shown by N and B single heteroatom-doped CBMs [76]. Moreover, S-doped graphene demonstrated an enhanced HER activity with an overpotential of 178 mV at a current density of 10 mA cm$^{-2}$ [77]. On the other hand, Chen et al. [78] found that N/S co-doped graphene could exhibit better HER performance, having an overpotential of 280 mV at a current density of 10 mA cm$^{-2}$ with a Tafel slope of 80.5 mV dec$^{-1}$. Another study found that N/P co-doped showed much lower overpotential and Tafel slope (420 mV at 10 mA cm$^{-2}$ and 91 mV dec$^{-1}$, respectively) than pristine graphene [26].

DFT calculations are performed by different researchers to elucidate the underlying mechanism of HER activity of CBMs. In theory, the HER pathway can be depicted as a three-state diagram containing an initial state of H$^+$ + e$^-$, an intermediate state of adsorbed H (as H* denotes an adsorption site), and a final state of ½ the H$_2$ product. Doping graphene with a heteroatom such as N can generate asymmetrical charge distributions on neighboring carbon atoms, resulting in bigger polarizations and increased affinities for H atoms [79]. Calculations using spin-unrestricted DFT found that interactions between H* and N-doped graphene were stronger than those between H* and graphene. Furthermore, doping with heteroatoms, for example, N atoms can affect the energy levels of the graphene matrix's valence orbitals. This can speed up the flow of electrons from graphene to catalytically active regions, resulting in the fast conversion of adsorbed H* species to molecular hydrogen [80]. Generally, a good hydrogen evolution catalyst should have a free energy of adsorbed H of approximately zero (*E$^0$), which can provide a fast proton/electron-transfer step as well as a fast hydrogen release process [81,82]. According to the computational results, pristine graphene has an endothermic DGH* of 1.82 eV, implying an energetically unfavorable interaction with hydrogen. The HER can barely proceed on pristine graphene because of the slow proton/electron transfer. Therefore, doping of graphene is commonly used. For example, graphitic N-doped graphene provides a large surface area and multi-electron transfer channel, which is exceptionally favorable for charge transport; in other words, HER supports achieving a DGH* of −0.55 eV, which is much lower than the Pt catalyst. The same goes for graphitic S-doped graphene which poses a DGH* of −0.3 eV. However, N, S co-doped graphene produces a synergistic effect that is more favorable to HER for both absorbing and desorbing the hydrogen atom, and it exhibits a DGH* of −0.12 eV, which is much close to the Pt catalyst (−0.09 eV) [78]. Therefore, the doping of CBMs has a great influence on their potential to work as catalytic materials; however, without knowing the fundamental of carbon doping chemistries, it is very tough to understand the overall activity of the materials. Additionally, in an N$_2$-saturated 0.5 M H$_2$SO$_4$ electrolyte, Jiang et al. [41] noted the exciting HER activity of the N-doped ultra-thin carbon nanosheets (NCN). Compared to the pristine carbon blocks and glass carbon (GC), the NCN samples show significantly increased HER activity. Surprisingly, NCN exhibits superior HER performance with about the same positive E$_{onset}$ as Pt/C (0.03 V), which is significantly more favorable than that of the other as-prepared NCN samples in our work. The obtained potential for NCN-1000-5 (−0.09 V) at a current density of 10 mA cm$^{-2}$ (Ej = 10) is only 51 mV more negative than that of Pt/C. The NCN's tiny Tafel slope of 43 mV dec$^{-1}$ is evidence of its good kinetic properties. The HER performance

of NCN is also superior to that of several other reported carbon materials. Similar LSV curves for NCN were obtained at sweep rates ranging from 5 to 100 mV s$^{-1}$, indicating the substance's strong stability for extremely active electrochemical processes. The nearly unchanged LSV curves after 500 CV cycles and the steady HER current after 12,000 s of continuous operation at $-0.15$ V provided additional evidence of the NCN's remarkable stability. (All data show in Table 3).

**Table 3.** Comparison of specific performance of the heteroatom-doped CBMs HER catalysts.

| | CBMs | Onset Potential (V) | Potential at 10 mAcm$^{-2}$ (V) | Tafel Slope (mVdec$^{-1}$) | Ref. |
|---|---|---|---|---|---|
| Single heteroatom doped CBMs | N-doped mesoporous graphene | $-0.15$ | $-0.24$ | 109 | [79] |
| | B-doped graphene | $-0.22$ | $-0.47$ | 99 | [83] |
| | O-doped CNTs | $-0.05$ | $-0.22$ | 71.3 | [75] |
| | N, S co-doped porous carbons | $-0.012$ | $-0.097$ | 57.4 | [84] |
| | N, S co-doped CNTs | $-0.05$ | $-0.12$ | 67.8 | [85] |
| Co-heteroatom doped CBMs | N, S co-doped nanoporous Graphene | $-0.14$ | $-0.39$ | 80.5 | [78] |
| | N, P co-doped nanoporous carbon | $-0.076$ | $-0.204$ | 58.4 | [86] |
| | N, P co-doped graphene | $-0.2$ | $-0.42$ | 91 | [26] |
| | N, P co-doped nanoporous Graphene | $-0.12$ | $-0.213$ | 79 | [87] |

All these electrocatalysts were tested in 0.5 M $H_2SO_4$ electrolyte solution.

*2.4. Methanol Oxidation*

2.4.1. Role of CBMs in Methanol Oxidation

Direct methanol fuel cells (DMFC) pose a promising track for the huge global demand for clean energy due to their low-cost, high-energy efficiency, low operation temperature, and easily transportable facilities in comparison with hydrogen fuel cells. The mechanism of DMFC comprises multiple elementary reactions that gradually end up with the final product of $CO_2$ [88]. The reaction detailed below involves a total transfer of six electrons and several possible intermediates in several reaction pathways: [89]

Anode Reaction: $CH_3OH + H_2O \rightarrow CO_2 + 6H^+ + 6e^-$

Cathode Reaction: $3/2\,O_2 + 6H^+ + 6\,e^- \rightarrow 2H_2O$

Overall Reaction: $CH_3OH + 3/2\,O_2 \rightarrow CO_2 + 2H_2O$

Although there are lots of complications that DMFCs are facing (e.g., high manufacturing cost, methanol crossover issues, low stability, and durability) that need to be addressed before successful commercialization. The sluggish reaction rate that occurs in the anode remains the main obstacle of DMFCs till now [90]. Electrocatalysts are mostly used to overcome these problems that contribute to the reliability of DMFCs, which is the key parameter to quantify the performance as well as the cost of the cell. In this case, various precious metals, such as platinum, rhodium, palladium, etc., and non-noble metals, such as nickel, copper, cobalt, etc., are being used as electrocatalysts [91].

Bagotski et al. presented the generally acknowledged premise of methanol oxidation via catalysis in 1977. He pointed out that methanol molecules are chemisorbed on the catalyst surface with the direct result of dehydrogenation forming $C^{***}-OH$ and $3H_{ads}$ (*** indicates three valence bonds with the surface). Step-by-step, partially dehydrogenated chemisorbed particles, such as $C^*H_2OH$, and $C^{**}HOH$, are formed in the middle of the dehydrogenation (Figure 7). Thus, the electro-oxidation of methanol on platinum proceeds via the interaction of adsorbed radicals $OH_{ads}$ with chemisorbed $C^*-OH$. Finally, $CO_2$ is produced with intermediate products such as $C^{**}=O$ and $OH-C^*=OH$. However, catalysts have their issues, such as platinum getting poisoned by CO, which is an intermediate product of the methanol oxidation reaction [92]. An advanced supporting material could be an effective solution to these types of problems which offers an increment of stability and durability, morphology, and amplified electrical conductivity. The intrinsic properties of CBMs have helped them to attain most of these qualities; however, they can be greatly increased by doping different heteroatoms.

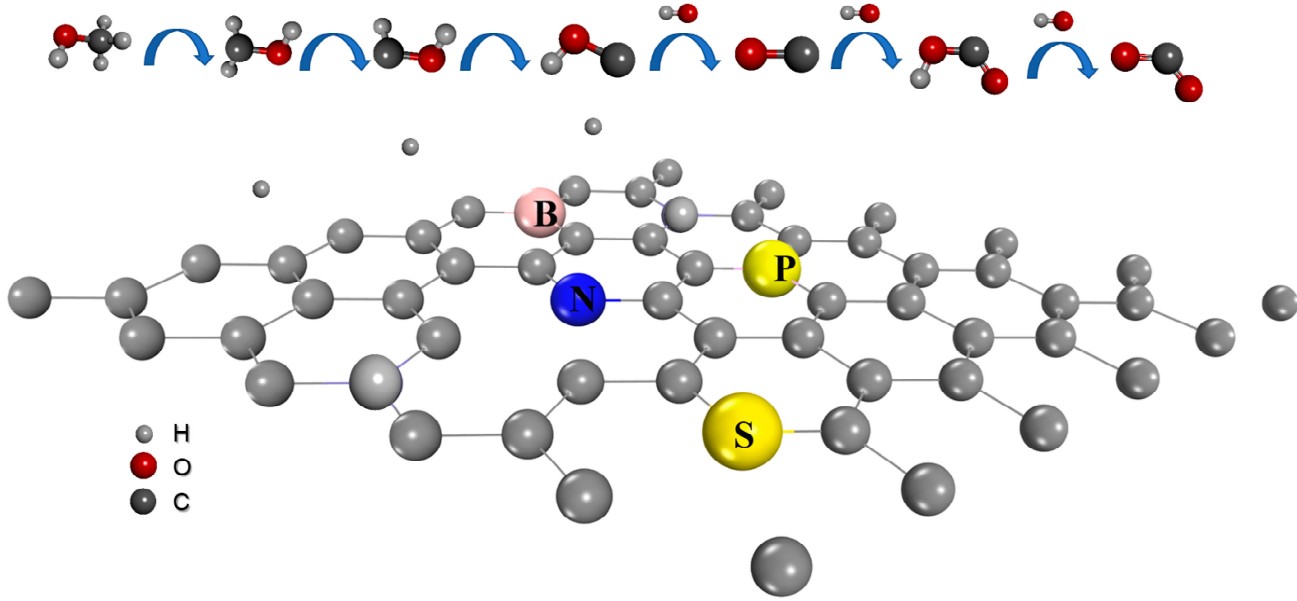

**Figure 7.** Mechanism of methanol oxidation in CBMs catalyst.

### 2.4.2. Effect of Heteroatom Doping on Methanol Oxidation

Different types of carbon-based support materials such as carbon black, carbon nanofiber, mesoporous carbon, carbon nanotubes, and graphene are being used as electrocatalysts for methanol-based fuel cells [91]. Doping of these CBMs with different heteroatoms would significantly alter various characteristics that have a distinct effect on the overall catalysis process as well as the enhancement of better electrochemical stability of the cell. For instance, nitrogen is the closest member of carbon on the periodic table and is considered one of the most significant dopant atoms of carbon-based materials for various applications. The doping of N atoms in CBMs can act as an active site that promotes the catalytic process of the reaction. Su et al. [93] reported that a platinum catalyst supported by an N-doped porous carbon nanosphere (PCN) could boost the methanol oxidation peak current to 343 mA mg$^{-1}$, which is significantly higher than the PCN-supported Pt of 297 mA mg$^{-1}$. Kou and Hsu (2010) reported that the N-doped porous carbon layer surrounding the CNTs-supported Pt hybrid (Pt/NC-25 CNT) received excellent electrocatalytic capabilities as Pt nanoparticles were properly embedded in the pores of CNTs, which boosted the methanol oxidation process significantly [94]. N-doped graphene-supported Pt-Au nanoparticles were also reported to enhance methanol oxidation catalytic activity appreciably (417 mA mg$^{-1}$), which is much higher than its undoped counterpart (186 mA mg$^{-1}$) [95]. B and S have been reported to have sounder dopant atom characteristics for application in the methanol oxidation reaction catalysis. B is affirmed to have strong interaction with the Pt nanoparticles hybridization between platinum d orbitals and boron p orbitals, indicating a direct bond between Pt and boron for a stable dispersion. B-doped CBMs are also reported to adsorb more oxygen which significantly improved the CO tolerance of Pt atoms [96]. Ahmadi et al. [97] used S as a dopant and found that the S-doped CNT-supported catalyst obtained an oxidation peak current of 862.8 mA mg$^{-1}$, which is much higher than the undoped catalyst (133.2 mA mg$^{-1}$). The difference between the electronegativity of S and C atoms could play an influential role in this case. Other non-metallic atoms such as O, Cl, Si, etc., and some metallic atoms such as Sn were also reported to have significantly improved the catalytic activity of methanol oxidation reaction. Furthermore, LV et al. studied the performance of an N-doped carbon nanotube-graphene hybrid nanostructure (NCNT-GHN) in which the graphene layers are distributed inside the CNT inner cavities and were designed to efficiently support noble metal (e.g., PtRu) nanoparticles with DMFC [98]. Figure 8 depicts the relationship between voltage and power density as measured at 30, 60, and 90 °C at various current densities.

Figure 8a–d shows that as the operating temperature rises from 30 to 90 °C, the output voltage and power density increase correspondingly. Peak power densities at 30, 60, and 90 °C, respectively, for the PtRu/NCNT-GHN catalyst are 195.3, 546.9, and 781.3 mA cm$^{-2}$. These values surpass both those of PtRu/CNT, which are 156.3, 429.7, and 664.1 mA cm$^{-2}$ at 30, 60, and 90 °C, respectively, and those of commercial catalysts by a wide margin. The results and their methanol oxidation activities are well-congruent.

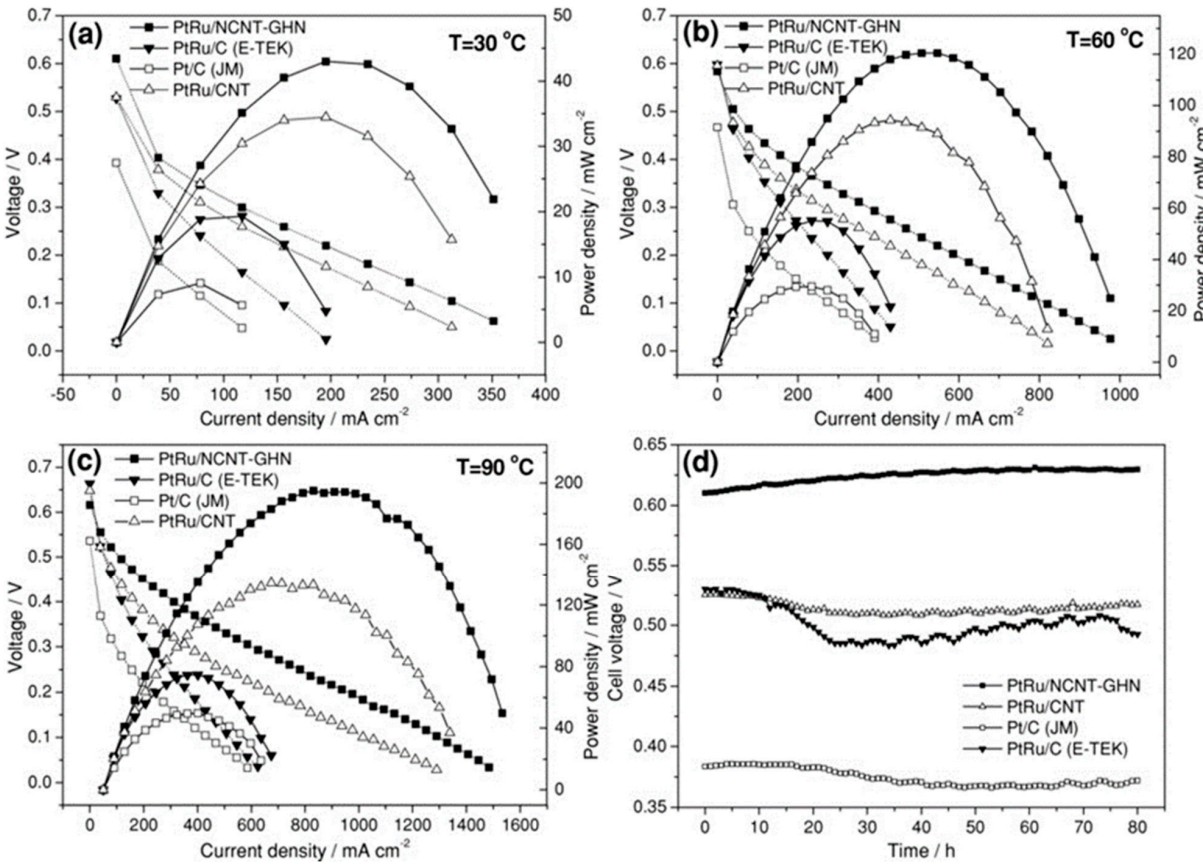

**Figure 8.** DMFC single-cell properties of different catalysts at (**a**) 30, (**b**) 60, and (**c**) 90 °C. (**d**) Durability test of DMFC single cells with different catalysts at 30 °C using 2 mol L$^{-1}$ methanol solution. It can be seen that the as-synthesized PtRu/NCNT-GHN catalyst demonstrates good stability at a higher output voltage than other MEO catalysts. Reproduced with permission from [98], Copyright@2011, John Wiley and Sons.

## 3. CBMs in Photocatalytic Decomposition of Organic Pollutants

### 3.1. Role of CBMs in the Photocatalysis Process

Water pollution is one of the most crucial global environmental problems these days. Amongst different water treatment methods, the photocatalytic degradation process has gained immense attention from researchers as it can remove different harmful organic pollutants by utilizing inexhaustible sunlight. However, a significant problem of this method is the requirement of various inorganic photocatalysts (e.g., TiO$_2$, ZnO, SnO$_2$ nanoparticles, etc.) which are toxic to many living species. On the other hand, carbon materials are relatively less toxic and environmentally friendly, and researchers found some CBMs performing outstandingly as catalysts for the photo-assisted degradation reactions of various harmful organic pollutants (e.g., different pharmaceutically active compounds, dyes, insecticides, etc.). Therefore, metal-free carbon-based photocatalysts are getting more attention day by day.

The involving mechanism of the photocatalytic process is quite simple. The carbon matrix, which acts as the catalyst, first adsorbs O$_2$ molecules from the atmosphere [99].

After obtaining a certain energy level from the irradiation, some exciton pairs (i.e., pairs of photo-induced electrons and holes) are generated. The photogenerated electrons, which have already undergone the "valence band to conduction band" transition, react with the adsorbed $O_2$ molecules and convert them into active $\cdot O_2{}^-$ species (Figure 9). In addition, the photogenerated protons may convert the hydroxyl ions of the water molecules to secondary reactive radical $\cdot OH$ [100]. The generated secondary reactive radicals then react with the pollutant molecules and degrade them to either lower molecular compounds or mineralize to $CO_2$ and $H_2O$ [101]. Other side reactions may also occur between the reactive $\cdot O_2{}^-$ species and the produced $H^+$ ions (reactions 20 to 26). The plausible degradation process can be described with the following reactions and is schematically shown in Figure 9 [100,102–104].

$$Photocatalyst \xrightarrow{h\vartheta} e^- + h^+ \tag{17}$$

$$h^+{}_{VB} + H_2O \rightarrow HO^\circ + H^+ \tag{18}$$

$$e^-{}_{CB} + O_2 \rightarrow O_2^- \tag{19}$$

$$O_2^- + H^+ \rightarrow HO_2{}^\circ \tag{20}$$

$$HO_2{}^\circ + HO_2{}^\circ \rightarrow H_2O_2 \tag{21}$$

$$O_2^- + HO_2{}^\circ \rightarrow O_2 + HO_2^- \tag{22}$$

$$HO_2^- + H^+ \rightarrow H_2O_2 \tag{23}$$

$$H_2O_2 + hv \rightarrow 2HO^\circ \tag{24}$$

$$H_2O_2 + O_2^- \rightarrow HO^\circ + OH^- + O_2 \tag{25}$$

$$e^-{}_{CB} + H_2O_2 \rightarrow HO^\circ + OH^- \tag{26}$$

$$O_2^- + Pollutant \rightarrow Degradation\ products \tag{27}$$

$$Or,\ HO^\circ + Pollutant \rightarrow Degradation\ products \tag{28}$$

$$Or,\ h^+ + Pollutant \rightarrow Degradation\ products \tag{29}$$

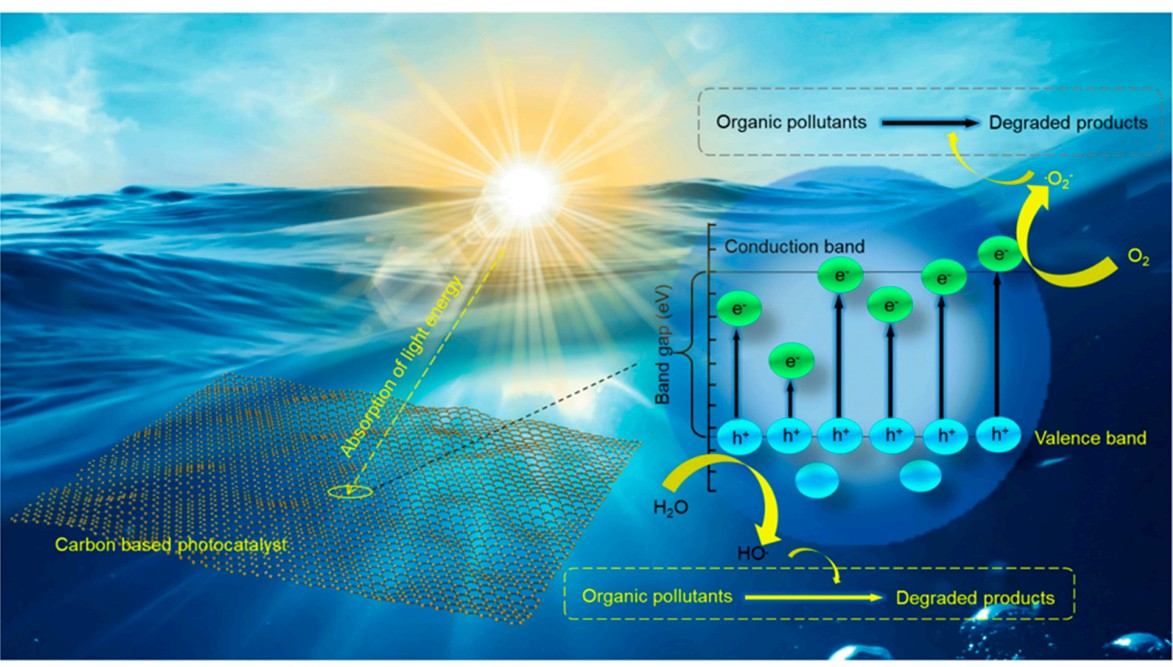

**Figure 9.** Mechanism of photocatalysis of organic pollutants by carbon-based materials.

The active catalytic species for the system (i.e., photogenerated secondary catalysts such as ·OH, $O_2^-$, $e^-$, or $h^+$) are evaluated by electron paramagnetic resonance and scavenger trapping experiments in the photocatalytic solution [99,105]. They may vary according to the chemical composition of the pollutants and the nature of the irradiation.

### 3.2. Research Progress of Heteroatom-Doped CBMs in Photocatalysis

Some advanced CBMs are promising candidates for the photocatalytic degradation reactions of various complex organic pollutants. However, one major drawback of these pure CBMs is their inherent low charge conductivity [106], which results in the recombination of some photogenerated electron/hole pairs. Consequently, the overall process efficiency is decreased. Different heteroatoms are doped with these pure CBMs to resolve this problem [100,107,108]. The dopant elements are composed of different electronic configurations compared to the regular *C*-atoms of the pristine CBMs. Therefore, *p*-type or *n*-type electronic properties can be induced into these materials by doping with various heteroatoms, and thus, the electronic conductivity of the pristine CBMs is improved. As a consequence, the rate of transfer of the photogenerated charge (electrons and holes) is increased, and the recombination of the electron/hole pairs is different [12]. Jourshabani et al. [107] described the effect of doping of mesoporous sulfur in graphitic carbon nitride (GCN) photocatalysts. The electrochemical impedance spectroscopy profile of the S-doped and undoped GCN showed a dramatic decrease in the charge transfer resistance of the photocatalysts after *S*-doping (Figure 10a). The delocalized lone pair electrons of the *S*-atom on the π-conjugated tri-s-triazine units were attributed to this extensive conductivity. A similar effect was observed in N-doped carbon nanosphere (CNS)-GCN composites [12]. Furthermore, the light absorption capacity of the doped CBMs is also increased due to the induced semiconducting properties [10]. Hu et al. [104] investigated the doping effect of phosphorous on the photocatalytic activity of mesoporous GCN. The catalytic activity of the pristine mesoporous GCN increased about 31.1 times after doping with phosphorous atoms. The synergistic effect of the higher light absorption in the visible range, faster separation, and conduction of photogenerated charges and the differed electron/hole pair recombination improves the catalytic performances of the doped CBMs [100,104,107]. However, the dopant concentration is an important factor to consider. Liu et al. [12] showed that the N-doping of a CNS-GCN composite decreases its electron transfer barrier. However, over dosage of dopant elements alleviates light absorption as well as resists electron transfer (Figure 10b).

The effect of heteroatom doping can also be explained by the band gap tuning principle of CBMs. For instance, Yan et al. [10] and Chai et al. [9] showed that the band gap of GCN (2.7 eV) is decreased to 2.66 eV after doping with boron, which was beneficial for its catalytic activity for pollutant degradation. Yan et al. [10] reported the band gap narrowing of GCN after boron doping at different temperatures. The fabrication temperature during doping may affect the surface morphology and, consequently, the absorption of light. Thereafter, the photocatalytic efficiency depends upon the doping temperature (Figure 10c). Liu et al. [96] reported that the self-narrowing of the band gap of sulfur and oxygen-co-doped carbon nitrides is responsible for the higher rate of photocatalytic decomposition of different organic neonicotinoids. The degradation efficiency of the system increased up to 57.6% by using the doped CBMs instead of the pristine pure carbon nitrides. A similar effect was also observed in another study, where the phosphorous doping in GCN prompted their catalytic activity for the mineralization of rhodamine B (RhB) and methyl orange dyes [108] (Figure 10d). In that study, the researchers obtained about 80–85% better degradation of methylene blue while using P-doped carbon nitrides instead of pure ones. Liu et al. [109] obtained about 65.2 times and 4.9 times higher degradation rates of methyl orange by using N-doped CNT/mpg-$C_3N_4$ as a photocatalyst than that of the pure CNTs and mpg-$C_3N_4$, respectively. So, the heteroatom-doped carbon-based catalysts can be a good option for the photocatalytic degradation reactions of organic pollutants, and they require more research attention soon. However, the presence of multiple organic pollutants in water bodies can be a major drawback of these catalysts for their practical

applications. Additionally, concentration variation of the pollutants in water, stability of surface morphology of the catalysts, temperature, and the pH of the contaminated solution as well as the presence of some inorganic oxidants may also affect the degradation efficiency of the photocatalysts [100,107]. Therefore, the development of a metal-free universal carbon-based catalyst is in great demand. Some outperforming works based on the doped CBMs have been summarized in Table 4.

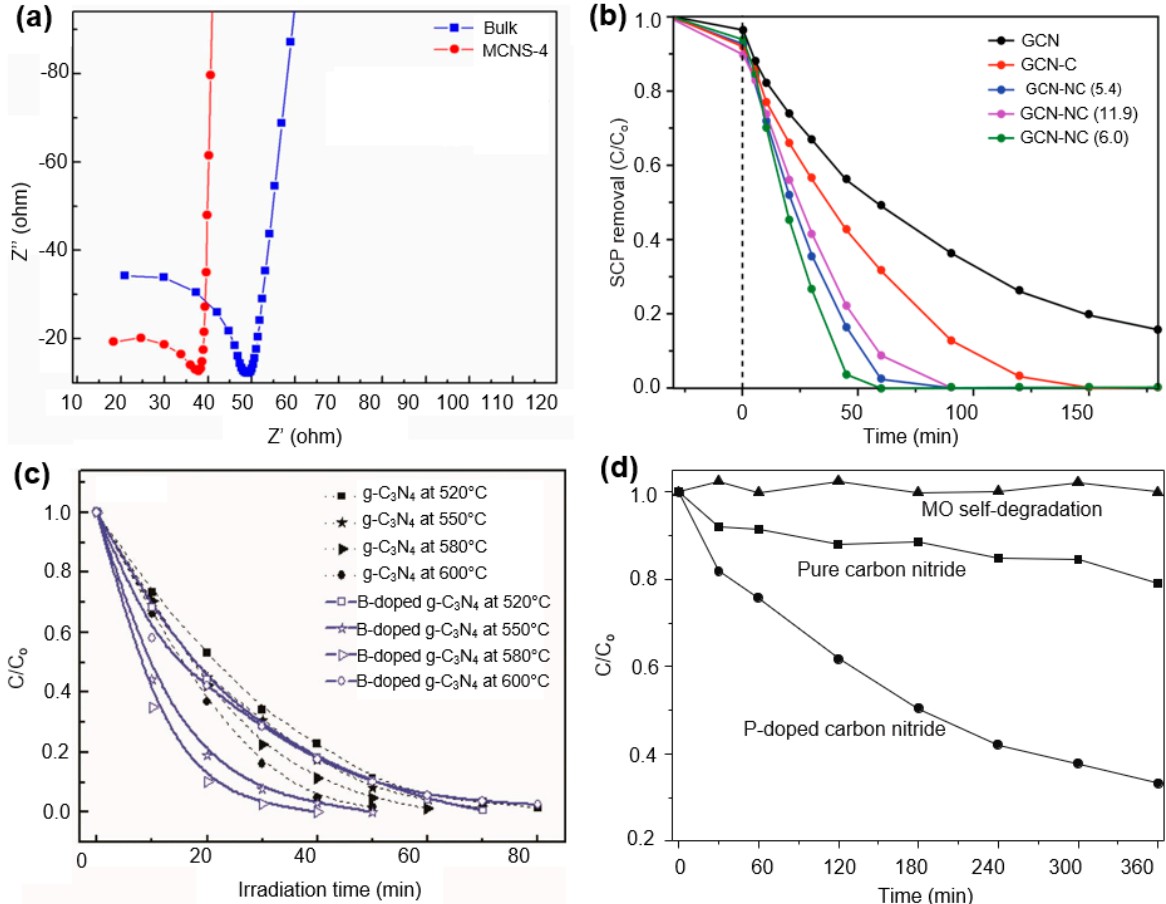

**Figure 10.** (**a**) The EIS profiles of bulk g-C$_3$N$_4$, and MCNS-4 samples. Reprinted with permission from Ref. [107], Copyright@ 2017, American Chemical Society. (**b**) Photocatalytic degradation of SCP over GCN, GCN-C, GCN-NC(5.4), GCN-NC(11.9), and GCN-NC(6.0) under visible light irradiation [12], reproduced under Creative Commons Attribution 4.0 International License; (**c**) Comparison of photocatalytic activities for RhB over g-C$_3$N$_4$ and B-doped g-C$_3$N$_4$ samples prepared at various temperatures; reprinted with permission from [10], Copyright@2010, American Chemical Society; (**d**) The photocatalytic degradation of MO over P-doped carbon nitride and pure carbon as a function of reaction time; reprinted with permission from [108], Elsevier Copyright Clearance Centre @2013.

**Table 4.** Photocatalytic performances of heteroatom-doped CBMs for pollutants degradation.

| Catalysts | Experimental Conditions (Catalyst Dosage, Pollutant conc. and Irradiation Source) | Degradation Efficiency | Ref. |
|---|---|---|---|
| Carbon dots (undoped) | 15 g/L catalyst<br>10 mg/L MB<br>10 mg/L RhB<br>310 W Hg-Xe lamp<br>with a UV cutoff filter (λ < 420 nm) | 90% MB in 60 min<br><br>50% RhB in 60 min | [110] |
| B-doped carbon dots | 2.5 mg/L MB<br>0.5 mg/L RhB<br>310 W Hg-Xe lamp<br>(UV-Visible light irradiation) | 99.9% MB in 170 min<br>99.9% RhB in 170 min | [111] |

<div align="center"><b>Table 4.</b> <i>Cont.</i></div>

| Catalysts | Experimental Conditions (Catalyst Dosage, Pollutant conc. and Irradiation Source) | Degradation Efficiency | Ref. |
|---|---|---|---|
| Carbon dots (undoped) | 0.001 g/L catalyst 10 mg/L RhB 500 W Xe lamp with a cutoff filter ($\lambda$ > 420 nm) | 0% in 60 min | [112] |
| Carbon dots/TNS | | 95% in 60 min | |
| Carbon dots (undoped) | 0.001 g/L catalyst $1 \times 10^{-5}$ mol/L RhB 500 W Xe lamp with a cutoff filter ($\lambda$ > 400 nm) | 0% in 30 min | [113] |
| Carbon dots/$TiO_2$ | | 95% in 30 min | |
| Pristine g-$C_3N_4$ (undoped) | 0.01 g/L catalyst 10 mg/L methyl orange 300 W halogen lamp ($\lambda$ > 400 nm) | 8.8% in 210 min | [107] |
| S doped g-$C_3N_4$ | | 82.7% in 210 min | |
| $MoS_2$/ZnS embedded N, S co-doped graphitic carbon | Dicofol pesticide Visible light irradiation lamp | 84.5% in 90 min | [105] |
| S, O co-doped g-$C_3N_4$ | 150 g/L catalyst 2 mg/L nitenpyram 300 W Xe lamp ($\lambda$ > 400 nm) | 91.4% in 30 min | [100] |
| P-doped carbon nitride ($HPCN_{0.5}$) | 2 mg/L dinotefuran 300 W halogen lamp ($\lambda$ > 400 nm) | 40.59% in 5 h | [102] |
| S doped rGO/S- g-$C_3N_4$/$Ag_3VO_3$ | 5 mg/L catalyst 10 mg/L methylene oxide 20 mg/L 2,4-dichlorophenoxy acetic acid (2,4-D) UV-Visible irradiation source ($\lambda$ = 464–664 nm) | 90.1% 2,4-D in 80 min 90.3% MO in 12 min | [114] |
| N,P co-doped carbon nanodots (CNDs)@ $TiO_2$ | 0.2 g/L catalyst 0.025 mmol/L 2,4-dichlorophenol Hg lamp with a cutoff filter ($\lambda$ < 400 nm) | 40% in 2 h | [101] |
| Exfoliated graphitic carbon nitride | 0.5 g/L catalyst 2 mg/L Bisphenol A 300 W Xe lamp ($\lambda$ > 420 nm) | 99.9% in 90 min | [115] |
| $TiO_2$@activated carbon | 1.2 g/L catalyst 100 mg/L phenol Natural sunlight | 100% in 120 min | [116] |
| N, P co-doped carbon quantum dots @ $TiO_2$ | 1 g/L catalyst 20 mg/L MB 300 W Xe lamp ($\lambda$ > 420 nm) | 100% in 15 min | [117] |
| S, N co-doped carbon quantum dots @ $TiO_2$ | 1 g/L catalyst 50 mg/L Acid red 88 Osram lamp (Visible light irradiation) | 77.29% in 120 min | [118] |
| Polyaniline @N-doped carbon nanodots (N/CNDs) | 0.5 g/L catalyst 0.1 g/L congo red White LED lamp (combination of $\lambda$ = 450 nm and 550 nm) | 100% in 20 min | [103] |
| P-doped mesoporous graphitic $C_3N_4$ | 0.4 g/L catalyst 25 mg/L Brilliant ponceau-5R 500 W Xe lamp with a cutoff filter ($\lambda$ > 420 nm) | 94.5% in 30 min | [104] |
| O-doped carbon nitride | 0.5 g/L catalyst 10 mg/L RhB 5 mg/L Tetracycline hydrochloride (TC-HCl) 300 W Xe lamp ($\lambda$ > 420 nm) | 95% RhB in 6 h 70% TC-HCl in 6 h | [119] |
| S-doped carbon nitride polymeric micro rods | 0.25 g/L catalyst $1 \times 10^{-5}$ mol/L RhB 300 W Xe lamp with a cutoff filter ($\lambda$ > 420 nm) | 97% in 15 min | [99] |
| N-doped CNT/mpg-$C_3N_4$ | 0.5 g/L catalyst 10 mg/L RhB 300 W Xe lamp with a cutoff filter ($\lambda$ > 400 nm) | 95% in 30 min | [109] |
| S-doped carbon quantum dots | 10 mg/L crystal violet 100 W UV-lamp ($\lambda$ = 395 nm) | 99.7% in 200 min | [120] |
| N, S co-doped carbon quantum dots @ ZnO | 0.4 g/L catalyst $2 \times 10^{-5}$ mol/L ciprofloxacin Natural sunlight | 85.8% in 50 min | [121] |
| N, P co-doped graphene quantum dots @ g-$C_3N_4$ | 1 g/L catalyst 10 mg/L MO 300 W Xe lamp ($\lambda$ > 420 nm) | 97.0% in 8 min | [122] |

**Table 4.** *Cont.*

| Catalysts | Experimental Conditions (Catalyst Dosage, Pollutant conc. and Irradiation Source) | Degradation Efficiency | Ref. |
|---|---|---|---|
| N-doped g-$C_3N_4$ | 1 g/L catalyst<br>Methylene blue<br>Sunlight | 90.0% in 3 h | [123] |
| N, S-doped carbon quantum dot-modified MIL-101(Fe) heterostructure | 0.4 g/L catalyst<br>Bisphenol A<br>Visible light irradiation | 100% in 60 min | [124] |

## 4. Conclusions and Future Perspectives

Catalysts are typically made of metals or metal oxides; however, heteroatom-doped carbon materials have many favorable circumstances over these more traditional catalysts, including earth-abundance, competitive prices, a high degree of specific surface areas and pore volumes, enormous quantities of surface defects, strong tolerance to acidic or alkaline environments, and structural tenability on both the morphological and molecular scales. Thus, they are superior replacements for metal catalysts. Because of the covalent chemical connections created between both the carbon and the heteroatom, most heteroatom-doped carbons have outstanding operational durability in contrast to metal alloys, which commonly struggle with separation difficulties. Because of their wide availability, low environmental impact, and absence of contamination by heavy metals, carbon catalysts are promising choices for green and sustainable chemistry. Overall, metal-free, heteroatom-doped carbon materials pose higher catalytic activity compared to pristine carbon materials. Moreover, this performance can be further enhanced through specific metal doping. Co-doping even results in better performance in catalytic applications. The introduction of heteroatoms into carbon structures brings about modifications to both the chemical and physical properties of those structures. As a consequence of these modifications, heteroatom-doped carbons perform far better in catalytic processes than un-doped carbon materials do.

The following are some of the upcoming difficulties and opportunities:

i. There is still a lack of understanding about the specific location of heteroatoms, the nature of their catalytic site in carbon materials and how they are doped. Combining experimental studies, state-of-the-art characterizations, and robust computational simulations is essential to a deeper understanding of the structure, mechanism, and thermodynamics of the catalytic core.

ii. While N-doped CBMS has been extensively researched as catalysts in several organic transformations, it is evident from this work that there are other dopants with suitable physicochemical properties, such as B, P, and S, that may be included in the carbon core. Additionally, future research should investigate the usage of heteroatoms, including tellurium, selenium, and others.

iii. Despite progress, controlling the number of heteroatoms, distributional homogeneity, bonding forms, and other aspects remains difficult. This is particularly true in situations where co-doping is involved. All of these factors influence the characteristics of doped carbons in catalytic processes, either directly or indirectly.

iv. It is still difficult to accurately adjust the location and quantity of heteroatoms in the carbon basal plane as well as to establish a clear connection between both the doped structure and catalytic performance.

v. Most of the research has focused on graphene and its derivatives (e.g., rGO, GCN, etc.) for evaluating the doping effects on their photocatalytic activity. The effect of heteroatom doping on other CBMs (such as CNT, activated carbon, and biochar) should also be studied.

vi. More research is required to develop universal carbon-based photocatalysts that will be capable of degrading multiple types of pollutants (e.g., dyes, pesticides, surfactants, etc.) present in bodies of water.

**Author Contributions:** J.A., M.S.H. and M.A. contributed equally and have written the manuscript, drafted the figures, and did all revisions. M.T.A., D.B., R.S., M.A.H. and M.Z. took part in manuscript writing, revision, and supervision and provided necessary support to improve the quality. M.B.A. and M.T.A. supervised J.A., M.S.H. and M.A.; did the online meeting, revision, and comments; and lead the team. M.B.A. did the conceptual design of this work. All authors have read and agreed to the published version of the manuscript.

**Funding:** This research received no external funding.

**Data Availability Statement:** Data for the figures can be shared upon request.

**Conflicts of Interest:** The authors declare no conflict of interest.

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
