# Peer review of "Catalytical Performance of Heteroatom Doped and Undoped Carbon-Based Materials"

_catalysts, doi:10.3390/catal13050823_

Round 1
Reviewer 1 Report
Please check the PDF file.

Author Response
We thank the editorial office for their constructive comments and suggestions to improve the manuscript’s quality. We have tried to address most of the concerns that were raised by editorial office. We have provided a separate response for each comment or question or suggestion. We have also used track change mode in the revised manuscript to show the necessary changes. We hope the revised manuscript will satisfy the reviewers and meet the journal standard.
Reviewer #1
General Comment: This review discusses the heteroatom doped carbon-based materials for catalytical applications, focusing mainly on energy and photocatalysis applications. I would like to recommend publication after addressing the following issues.
Response: We thank the reviewer for the constructive comments and for the suggestion.
Specific comments:
Comment 1: The abstract is too generalized, it is difficult to understand the scope by reading it. Please check the language and grammar as well, use more scientific words. This is a review article, avoid using word “results” at the abstract.
Response: Thanks to the reviewer for finding out this issue in abstract. According to the suggestions of this reviewer, we have improved the grammatical issues, and changed language using the more scientific words. Also, the scope of this work is now clearly addressed. Finally, we have also avoided the results word from the abstract.
Comment 2: The whole manuscript needs grammatical corrections.
Response: We have revised the whole manuscript and avoided any grammatical issues.
Comment 3: Provide all figures with higher resolutions.
Response: We have separately provided high resolution images in PowerPoints slides. Anyway, we have updated it again.
Comment 4: Figure 1 resolution need to be improved, keep similar figure size throughout the manuscript, the pathway or step number in the figure would help new readers to understand the mechanism, it is quite confusing.
Response: Figure 1 resolution has been improved in the revised manuscript.
Comment 5: Line 137-138, what do you mean by volatile metal?
Response: It was a typo mistake. We have updated the information as “volatile organic compounds”.
Comment 6: Line 176-178, “N-P co-doped graphene had a minimum OER over the potential of 390 mV” this sentence doesn’t make sense due to typos. Volcano plot is not explained well.
Response: Thanks for finding out this point. The correct sentence would be “N-P co-doped graphene catalyst had a minimum OER overpotential of 390 mV, lower than the best catalyst identified theoretically (420 mV for OER in RuO2).” We have also updated the term “over the potential” to overpotential.
Reviewer 2 Report
This work pay attention to the function of heteroatom-based doped and undoped CBMs for catalytical applications together with a discussion about the future perspectives of doped and undoped CBMs. This provides valuable information for future designs with high catalytic performance of doped and undoped CBMs. However, there is less summary of recent publications, especially research articles published in 2021 and 2022, and improvements are recommended. In addition, the author should provide a summary diagram of the advantages, current situation, improvement methods and future development trends of doped and undoped CBMs in catalytical applications to help readers better understand.
Author Response
We thank the editorial office for their constructive comments and suggestions to improve the manuscript’s quality. We have tried to address most of the concerns that were raised by editorial office. We have provided a separate response for each comment or question or suggestion. We have also used track change mode in the revised manuscript to show the necessary changes. We hope the revised manuscript will satisfy the reviewers and meet the journal standard.
Reviewer #2
General Comment: This work pays attention to the function of heteroatom-based doped and undoped CBMs for catalytical applications together with a discussion about the future perspectives of doped and undoped CBMs. This provides valuable information for future designs with high catalytic performance of doped and undoped CBMs.
Response: Thanks for the summary statement.
Comment 1: There is less summary of recent publications, especially research articles published in 2021 and 2022, and improvements are recommended.
Response: According to the suggestion of this reviewer, we have updated many references from recent publications i.e., 2021-2023 in the revised manuscript. The relevant references are 1, 2, 3, 5, 17, 19, 30, 88, 98, 99, 111, 116, 118, 119, and 120.
Comment 2: In addition, the author should provide a summary diagram of the advantages, current situation, improvement methods and future development trends of doped and undoped CBMs in catalytical applications to help readers better understand.
Response: According to the suggestion of this reviewer, we have added one new figure in the revised manuscript.
Reviewer 3 Report
In this review, the authors introduced the function of heteroatom-based doped and undoped carbon-based materials for catalytical applications together with a discussion about the underlying chemistries of catalysis. The review can be published in "catalysts" once the following issues can be adequate addressed.
1. There are published reviews for carbon-based materials. In the introduction, the authors must describe the difference of this review from previously published ones. The authors must describe why there is still room to write another review in this regard.
2. After describing the section about “Effect of hetero-atom doping on each application”, one section should be added which briefly describes the research progress of carbon-based materials.
3. It is better to add subheadings to introduce the effect of hetero-atom doping on each application.
4. Please change “Table 4”to a three-line table.
5. The radicals from the reactions (19-28) should be more professional, such as change “°O2−” to “O2-”.
6. The words and grammars of the manuscript should be carefully checked.
Author Response
We thank the editorial office for their constructive comments and suggestions to improve the manuscript’s quality. We have tried to address most of the concerns that were raised by editorial office. We have provided a separate response for each comment or question or suggestion. We have also used track change mode in the revised manuscript to show the necessary changes. We hope the revised manuscript will satisfy the reviewers and meet the journal standard.
Reviewer #3
General Comment: In this review, the authors introduced the function of heteroatom-based doped and undoped carbon-based materials for catalytical applications together with a discussion about the underlying chemistries of catalysis. The review can be published in "catalysts" once the following issues can be adequate addressed.
Response: Thanks for the overall comments and for the suggestions to improve the manuscript quality.
Comment 1: There are published reviews for carbon-based materials. In the introduction, the authors must describe the difference of this review from previously published ones. The authors must describe why there is still room to write another review in this regard.
Response: Thanks. We have now clearly described the study gap and motive of our article in the introduction part (line no. 84-107 in the edited manuscript):
“Although some existing reviews are available on the photocatalytic and electrocatalytic application of various doped CBMs, most of them are based on some specific CBMs (e.g., carbon dots, graphene), specific element/s doping (e.g., doing with nitrogen), or some specific reactions (e.g., HER, ORR). For instance, recently, Salinas-Torres et al., [13] reviewed the application of nitrogen-doped CBMs for hydrogen generation reactions. But, the effect of other heteroatoms doping was not discussed in that article. Zhang et al., [14] studied the recent advances in the application of doped carbon dots for electrocatalytic reactions. However, the study was confined to carbon dots only. Other heteroatom-doped CBMs such as graphene, GCN, and CNTs were not reviewed. Hu et al., [15] reviewed the doped CBMs for various electrocatalysis reactions. The photocatalytic application of such materials was not discussed in that study. Other review articles based on the N-doped CBMs for ORR [16], transition metals doped CBMs for HER [17], catalytic applications of metal-N-doped CBMs [18], and electrocatalytic applications of non-N-doped CBMs [19] are also available. However, to the best of the authors’ knowledge, there is no such review work in literature where the role and underlying chemistry of heteroatom doping on the electrocatalytic and photocatalytic behavior of different CBMs (e.g., graphene, carbon dots, GCN, CNTs etc.) has been discussed together.
In our study, we have reviewed the photocatalytic and electrocatalytic application of various heteroatoms doped CBMs in the reactions such as hydrogel evolution reaction (HER), oxygen evolution reaction (OER), ORR, methanol oxidation and pollutants catalysis. In addition, the effects of various dopant elements on the performance of these CBMs and the underlying chemistries of these effects have also been discussed in detail.”
Comment 2: After describing the section about “Effect of hetero-atom doping on each application”, one section should be added which briefly describes the research progress of carbon-based materials. It is better to add subheadings to introduce the effect of hetero-atom doping on each application.
Response: We have divided our main headings into different subheadings where “effect of hetero-atom doping on each applications” where described clearly with the performance. So we want to keep the current subheadings as it looks more logical to us. Hope reviewer will understand it.
Comment 3: Please change “Table 4” to a three-line table.
Response: Thanks for the suggestion. The “Table 4” has been changed accordingly.
Comment 4: The radicals from the reactions (19-28) should be more professional, such as change “°O2−” to “O2-”.
Response. The equations have been changed accordingly.
Comment 5: The words and grammars of the manuscript should be carefully checked.
Response: We have revised the whole manuscript and checked again to avoid all grammar issues.
Round 2
Reviewer 2 Report
The manuscript now can be accepted.Reviewer 3 Report
The quality of the manuscript has been greatly improved after revision, and I recommend the manuscript to be published in Journal of catalysts.